# Upregulation of Transferrin and Major Royal Jelly Proteins in the Spermathecal Fluid of Mated Honeybee (*Apis mellifera*) Queens

**DOI:** 10.3390/insects12080690

**Published:** 2021-07-31

**Authors:** Hee-Geun Park, Bo-Yeon Kim, Jin-Myung Kim, Yong-Soo Choi, Hyung-Joo Yoon, Kwang-Sik Lee, Byung-Rae Jin

**Affiliations:** 1College of Natural Resources and Life Science, Dong-A University, Busan 49315, Korea; phg1314@naver.com (H.-G.P.); boyeon@dau.ac.kr (B.-Y.K.); kjm8578@naver.com (J.-M.K.); 2Department of Agricultural Biology, National Academy of Agricultural Science, Wanju 55365, Korea; beechoi@korea.kr (Y.-S.C.); yoonhj1023@korea.kr (H.-J.Y.)

**Keywords:** *Apis mellifera*, major royal jelly protein, transferrin, spermatheca, spermathecal fluid

## Abstract

**Simple Summary:**

To understand the mechanisms underlying long-term storage and survival of sperm in honeybee *Apis mellifera* queens, previous studies have elucidated the components of honeybee spermathecal fluid. However, the expression profiles of transferrin (Tf) and major royal jelly proteins 1–9 (MRJPs 1–9) in the spermatheca and spermathecal fluid of mated honeybee queens have still not been characterized. In this study, we confirmed upregulation of Tf and MRJPs in the spermatheca and spermathecal fluid of mated honeybee queens by using RNA sequencing, reverse transcription-polymerase chain reaction, and Western blot analyses. The levels of Tf and antioxidant enzymes were elevated in the spermathecal fluid of the mated queens, paralleling the levels of reactive oxygen species, H_2_O_2_, and iron. The increased levels of MRJPs, especially MRJP1, MRJP4, and MRJP6, in the spermathecal fluid of mated queens may be responsible for energy provision during sperm storage in honeybee queens. Overall, our findings indicate that Tf and MRJPs are upregulated in the spermatheca and spermathecal fluid of mated honeybee queens, providing a novel insight into antioxidant defense and energy metabolism for stored sperm in honeybee queens.

**Abstract:**

Sperm storage in the spermathecae of honeybee (*Apis mellifera*) queens is vital for reproduction of honeybees. However, the molecular mechanisms whereby queens store sperm in a viable state over prolonged periods in the spermatheca are not fully understood. Here, we conducted RNA sequencing analysis of the spermathecae in both virgin and mated *A. mellifera* queens 24 h after mating and observed that the genes encoding transferrin (Tf) and major royal jelly proteins (MRJPs) were differentially expressed in the spermathecae of mated queens. The concentrations of Tf and antioxidant proteins such as superoxide dismutase 1, catalase, and glutathione peroxidase as well as the levels of reactive oxygen species, H_2_O_2_, and iron were higher in the spermathecal fluid of the mated queens than in virgin queens. Tf upregulation is likely to perform a protective role against the Fenton reaction occurring between iron and H_2_O_2_ in the antioxidant pathway in the mated queen’s spermathecal fluid. Furthermore, MRJPs—especially MRJP1, MRJP4, and MRJP6—were upregulated in the mated queen’s spermathecal fluid, indicating that they may serve as antimicrobial and antioxidant agents as well as an energy source for stored sperm in the spermathecal fluid of honeybee queens. Together, our findings show that Tf and MRJPs are upregulated in the spermatheca and spermathecal fluid of mated honeybee queens.

## 1. Introduction

Honeybee (*Apis mellifera*) queens mate during the early stages of their lives, store sperm in the spermatheca, and subsequently use the stored sperm throughout their lifetimes [1,2]. Sperm storage in the spermatheca of honeybee queens plays a vital role in the reproduction of honeybees, but the molecular mechanisms whereby queens can maintain sperm in a viable state in the spermatheca over prolonged periods are still not completely understood. Spermathecal fluid, which is secreted by the spermathecal glands of honeybee queens, contains proteins and metabolites [2,3,4,5] and has been shown to facilitate long-term sperm storage [2,4,6,7]. Several proteins, such as the glycolytic enzyme triosephosphate isomerase [3] and antioxidant enzymes [8,9,10], have been proposed to be responsible for sperm viability in the spermathecal fluid of honeybee queens. 

Extensive studies, including proteomic, metabolomic, and transcriptomic analyses, have been conducted on the storage and survival of sperm in *A. mellifera* [1,2,4,5,10,11,12,13,14]. A previous study identified >100 proteins as major spermathecal fluid constituents in *A. mellifera* queens and analyzed the association of spermathecal fluid proteins with energy metabolism and antioxidant defense [2]. Recent studies have identified >10,000 transcripts and 2778 proteins in the mated or virgin *A. mellifera* queen’s spermathecae via transcriptomic and proteomic analyses, respectively [13,14]. The results from the transcriptomic analyses revealed the gene expression patterns during sperm storage after mating and showed that Kielin/chordin-like and trehalase transcripts are highly expressed in the spermathecae of the mated queens, in comparison with their levels in virgin queens [14]. The glycolytic metabolite glyceraldehyde-3-phosphate reportedly facilitates the long-term survival and energy (ATP) production of honeybee sperm during their storage in the spermathecae, because this metabolite functions independently of oxygen and prevents sperm damage caused by reactive oxygen species (ROS) [4]. Moreover, the activities of catalase and glutathione S-transferase (GST) in the mated *A. mellifera* queen’s spermathecae were higher than those in the virgin queen’s spermathecae [8]. The upregulation of antioxidant genes such as *catalase*, *thioredoxin 2*, and *thioredoxin reductase 1* after mating is associated with sperm viability within the mated *A. mellifera* queen’s spermathecae [10]. Moreover, stored sperm showed a reduced metabolic rate and lower ROS levels [15].

The metabolic energy profiles of sperm and spermathecae were characterized on the basis of the activities of enzymes involved in the aerobic and anaerobic carbohydrate metabolisms during the life cycles of honeybee queens [11]. Glycolytic enzymes such as glyceraldehyde-3-phosphate dehydrogenase (GAPDH) and enolase are abundant in the stored sperm [16] and spermathecal fluid of honeybee queens [2]. Heat-induced changes in GAPDH expression influence the viability of sperm in the spermathecae of honeybee queens [13]. Glycolytic enzymes in the spermathecal fluid of honeybee queens have been linked to sperm energy metabolism [4], whereas antioxidant enzymes reportedly protect the stored sperm against ROS activity [8,9,10].

Previous studies have investigated the energy metabolism and antioxidant defense during sperm storage in the spermathecae of honeybee queens [4.15]. Moreover, honeybee spermathecal fluid components have been elucidated to clarify the mechanisms underlying long-term storage and survival of sperm in the spermathecae of *A. mellifera* queens [1,2]. Proteomic analysis has also demonstrated that the spermathecal fluid of *A. mellifera* queens contains transferrin (Tf) and two major royal jelly proteins (MRJPs 8 and 9) [2]. However, the expression patterns and roles of Tf and MRJPs 1–9 in the spermathecae of mated honeybee queens have still not been described in detail. Moreover, the expression patterns of MRJPs 1–9 in the spermathecal fluid of mated queens have not been characterized, and the potential roles of increased levels of Tf in protecting the sperm from ROS action remain unclear. Therefore, characterization of the expression patterns of MRJPs and Tf in the spermathecal fluid of mated queens is essential to understand the mechanisms underlying sperm storage in honeybee queens.

In this study, we aimed to address these questions by using RNA sequencing (RNA-seq) analyses to compare the spermathecae of virgin and mated *A. mellifera* queens 24 h after mating. We found that unlike the spermathecae of virgin queens, the spermathecae of mated queens showed upregulation of Tf, which functions as an antioxidant protein with a protective role against the Fenton reaction occurring between Fe^2+^ and H_2_O_2_ [17,18,19,20,21,22], while the mated queen’s spermathecal fluid showed increased levels of ROS, H_2_O_2_, and iron. We also showed that MRJPs 1–9—especially MRJP1, MRJP4, and MRJP6—are upregulated in the spermathecae and spermathecal fluid of mated queens. These results suggest that upregulation of MRJPs and Tf may be involved in energy metabolism and antioxidant defense during sperm storage in the mated queen’s spermathecal fluid. 

## 2. Materials and Methods

### 2.1. Sample Preparation

The bees used for the experiments were reared at Dong-A University. To produce honeybee queens, one-day-old larvae were grafted into queen cells, placed on frames, and introduced into a de-queened colony [23]. Virgin queens were kept in nucleus colonies, and six-day-old virgin queens were artificially inseminated using a previously developed method to produce mated queens [24]. Spermathecae from six-day-old virgin queens after emergence were dissected on ice by using a stereomicroscope (Zeiss, Jena, Germany), while spermathecae from mated queens were dissected 24 h after artificial insemination. The spermathecae were punctured to extract the spermathecal fluid, which was subsequently centrifuged at 14,000× *g* for 10 min to remove tissue debris and sperm (in the fluid extracted from mated queens). The spermathecae of virgin and mated queens and the sperm in the spermathecal fluid of mated queens were observed using a stereomicroscope. The supernatant (spermathecal fluid) was stored at −70 °C until further use. Spermathecal samples, including the spermathecal glands from virgin queens (without spermathecal fluid) and mated queens (without spermathecal fluid and sperm), were collected and washed with phosphate-buffered saline (PBS: 140 mM NaCl, 27 mM KCl, 8 mM Na_2_HPO_4_, and 1.5 mM KH_2_PO_4_, pH 7.4) for subsequent molecular analyses. All samples were collected from three biological replicates: three pooled spermatheca or spermathecal fluid samples (n = 9 virgin or mated queens).

### 2.2. RNA Sequencing and Data Analysis

Total RNA was isolated from the spermatheca samples (without spermathecal fluid or sperm) from virgin or mated queens by using TRIzol reagent (Invitrogen, Carlsbad, CA, USA) according to the manufacturer’s protocol. The RNA samples for RNA sequencing were prepared from three biological replicates: three pooled spermatheca samples (n = 9 virgin or mated queens). RNA sequencing was performed using the NovaSeq 6000 Sequencing System (Illumina, San Diego, CA, USA) at Macrogen Inc. (Seoul, Korea). Briefly, sequence libraries were constructed using the TruSeq RNA Library Preparation Kit (Illumina) by following the manufacturer’s protocol. The sequences were qualified using both the FastQC (www.bioinformatics.babraham.ac.uk/projects/fastqc, accessed on 30 July 2021) and Trimmomatic programs (www.usadellab.org/cms/index/php?page=trimmomatic, accessed on 30 July 2021). RNA reads were mapped to a reference genome (GCF_000002195.4_Amel_4.5) [14] by using the HISAT2 aligner [25]. Transcript counts, representing gene expression levels, were determined as fragments per kilobase of exon per million fragments mapped (FPKM). The analysis of differentially expressed genes (DEGs) was performed using the Database for Annotation, Visualization and Integrated Discovery (DAVID) platform [26] (https://david.ncifcrf.gov/home.jsp, accessed on 30 July 2021), and log_2_ values were calculated and normalized through quantile normalization. Fold changes were calculated as the mean log_2_ (FPKM+1) value, and an independent *t*-test was used to determine the statistical significance of the differences between the values obtained from mated and virgin queens. Differences with *p*-values < 0.05 were considered statistically significant. 

### 2.3. Reverse Transcription-PCR (RT-PCR)

To examine the expression patterns of genes encoding antioxidant proteins and MRJPs in the spermathecae, the cDNAs for catalase (GenBank accession number GB41427), glutathione peroxidase (GTPX) (GenBank accession number GB47478), transferrin (Tf) (GenBank accession number GB50226), superoxide dismutase 1 (SOD1) (GenBank accession number GB47880), and MRJPs (MRJP1, MRJP2, MRJP3, MRJP4, MRJP5, MRJP6, MRJP7, MRJP8, and MRJP9) were amplified from the total RNA by using RT-PCR. The PCR primers for *MRJPs 1–9* have been described in our previous studies [27,28]: *MRJP1* (GenBank accession number NM_001011579), *MRJP2* (NM_001011580), *MRJP3* (NM_001011601), *MRJP4* (NM_001011610), *MRJP5* (NM_001011599), *MRJP6* (NM_001011622), *MRJP7* (NM_001014429), *MRJP8* (NM_001011564), and *MRJP9* (NM_001024697). The PCR primer sets used in this study were designed to amplify full sequences and are listed in Table 1. *β–Actin* (GenBank accession number NM_001185145) was used as the internal control. PCR was performed using a thermocycler (Takara Bio, Otsu, Japan) as follows: 94 °C for 2 min, 35 cycles of amplification (94 °C for 1 min, 53 °C for 1 min, and 72 °C for 2 min), and 72 °C for 10 min. The resulting fragments were observed by electrophoresis with a 1.0% agarose gel, and the cDNA concentrations were measured [29]. Finally, the resulting fragments were verified by DNA sequence analysis using a 3730XL DNA Analyzer (Applied Biosystems, Foster City, CA, USA). 

### 2.4. Recombinant Protein Expression and Purification

Recombinant catalase—GTPX, Tf, and SOD1—were produced using a baculovirus expression system [30]. The protein-coding sequences of catalase, GTPX, Tf, and SOD1 were PCR-amplified, and a His-tag sequence was included in each cDNA sequence. The cDNA fragments corresponding to the protein sequences were inserted into the *pBacPAK8* transfer vector (Clontech, Palo Alto, CA, USA), in which gene expression is under the control of the polyhedrin promoter of *Autographa californica* nucleopolyhedrovirus (AcNPV). For the expression experiments, 500 ng of the constructs and 100 ng of AcNPV viral DNA [30] were co-transfected into 1.0–1.5 × 10^6^ *Spodoptera frugiperda* (Sf9) cells for 5 h by using Lipofectin transfection reagent (Gibco BRL, Gaithersburg, MD, USA). The transfected cells were cultured in TC100 medium (Gibco BRL) with 10% fetal bovine serum (FBS, Gibco BRL) at 27 °C for 5 d. Recombinant baculoviruses were propagated in Sf9 cells cultured in TC100 medium at 27 °C to produce the recombinant proteins, which were then purified using the MagneHis Protein Purification System (Promega, Madison, WI, USA). Protein concentrations were determined using the Bio-Rad Protein Assay Kit (Bio-Rad, Hercules, CA, USA). The recombinant MRJP proteins (MRJPs 1–9) produced in baculovirus-infected insect cells were used as described in our previous studies [27,28].

### 2.5. Polyclonal Antibody Production and Western Blotting 

Purified recombinant proteins, such as catalase, GTPX, Tf, and SOD1, were mixed with an equal volume of Freund’s complete adjuvant (a total of 200 μL) and injected into BALB/c mice (Samtako Bio Korea Co., Osan, Korea). Two successive injections with a one-week interval were administered using antigens mixed with equal volumes of Freund’s incomplete adjuvant (a total of 200 μL). Blood was collected 3 d after the second injection and centrifuged at 10,000× *g* for 5 min. The supernatants were used as the antibodies for the Western blot analyses. The anti-MRJP 1–9 antibodies raised against recombinant MRJPs 1–9 were used as described in our previous studies [27,28].

Western blot analysis was performed using an enhanced chemiluminescence (ECL) western blot system (Amersham Biosciences, Piscataway, NJ, USA). Protein samples were mixed with the sample buffer (0.0625 M Tri-HCl, pH 6.8, 2% sodium dodecyl sulfate [SDS], 10% glycerol, 5% β-mercaptoethanol, and 0.125% bromophenol blue) and then boiled for 5 min. The protein samples (5 μg/lane) from the spermathecae (without spermathecal fluid or sperm) and the spermathecal fluid (without sperm) of virgin and mated queens were separated using SDS-polyacrylamide gel electrophoresis (SDS-PAGE) with a 12% gel. After the electrophoresis, the proteins were transferred onto a nitrocellulose membrane (Schleicher & Schuell, Dassel, Germany), which was subsequently blocked with 1% bovine serum albumin. The membranes were then incubated with the antiserum solution (1:1000 *v*/*v*) at room temperature for 1 h and washed in TBST (10 mM Tri-HCl, pH 8.0, 100 mM NaCl, and 0.005% (*v*/*v*) Tween 20). They were incubated with a 1:5000 (*v*/*v*) diluted anti-mouse IgG horseradish peroxidase (HRP) conjugate and HRP–streptavidin complex. After repeated washes with TBST, the membranes were incubated with ECL detection reagents (Amersham Biosciences) and exposed to an autoradiography film. The western blot images were analyzed by determining the integrated density of each band area by using a computerized image analysis system (Alpha Innotech Co., San Leandro, CA, USA) and AlphaEaseFC (ver. 4.0). The relative levels of each protein are shown as the mean values from three measurements, which were calculated relative to the levels in the spermathecal fluid of virgin queens (shown as 100%).

### 2.6. Measurement of ROS and Iron Levels

The ROS, H_2_O_2_, and iron levels in the spermathecal fluid of virgin and mated honeybee queens were quantified using the OxiSelect In Vitro ROS/RNS Assay Kit (Green Fluorescence; Cell Biolabs, Inc., San Diego, CA, USA), Hydrogen Peroxide Assay Kit (Abcam, Cambridge, UK), and Iron Colorimetric Assay Kit (BioVision Inc., Milpitas, CA, USA) according to the instructions of the manufacturers. Experiments were performed three times by using independent samples.

### 2.7. Enzymatic Activity Assay

The catalase, peroxidase, and SOD enzyme activities in the spermathecal fluid were assayed using a Catalase Activity Colorimetric/Fluorometric Assay Kit (BioVision Inc.), Peroxidase Activity Colorimetric/Fluorometric Assay Kit (BioVision Inc.), and SOD Activity Assay Kit (BioVision Inc.) according to the manufacturer’s instructions.

### 2.8. Statistical Analysis 

Data are presented as the mean ± SD. To determine the statistical significance of the differences in values between the mated and virgin queens, unpaired two-tailed Student’s *t*-test and the statistical software SPSS version 22.0 (IBM Inc., Chicago, IL, USA) were used. We set statistical significance at *p*-values of <0.05, <0.01, and <0.001.

## 3. Results

### 3.1. Identification of Genes Differentially Expressed between the Mated and Virgin Queens’ Spermathecae 

We collected spermathecae and spermathecal fluid from mated queens 24 h after mating and compared them with those obtained from virgin queens (Figure 1). At 24 h after mating, the color of the spermathecae of mated queens changed, and sperms were observed in their spermathecal fluid. 

The spermathecal fluid was collected from the dissected spermathecae of mated queens and virgin queens (Figure 1); subsequently, the spermathecal tissues, including the spermathecal glands, were used for total RNA extraction. Using RNA-seq analysis, we compared the transcriptome profiles of the spermathecae of mated queens with those of the virgins. We identified genes that were differentially expressed in the spermathecae of mated queens. A total of 13,064 genes showed differential expression between the mated and virgin queens’ spermathecae. Of these, 749 genes were considered significant with an independent *t*-test *p* < 0.05 and a mated queen/virgin queen expression ratio >2.0 (Appendix A). A list of the selected genes that encode antioxidant proteins and MRJPs is shown in Table 2. Of these genes encoding antioxidant proteins, *catalase, GTPX*, and *Tf* were upregulated in the spermathecae of the mated queens (Table 2), whereas *SOD1* was not differentially expressed. Moreover, *MRJPs*, especially *MRJP1*, *MRJP4*, and *MRJP6*, were upregulated in the spermathecae of the mated queens in comparison with their levels in the virgin queens (Table 2). We focused on MRJPs and the antioxidant proteins SOD1, catalase, GTPX, and Tf for further investigation. 

### 3.2. Antioxidant Defense in the Spermathecae of Mated Queens and Virgin Queens

We found that Tf as well as catalase and GTPX were differentially expressed in the spermathecae of the mated queens. Therefore, we examined the transcript levels of the antioxidant proteins catalase, GTPX, and Tf in the spermathecae of mated and virgin queens (Figure 2A). Considering the antioxidant pathway, SOD1, which catalyzes the dismutation of superoxide to H_2_O_2_ [31,32], was examined in this study (Figure 2A). We confirmed that *catalase*, *GTPX*, and *Tf* transcript levels are upregulated in the mated queen’s spermathecae in comparison with the levels in the virgin queens; however, the *SOD1* transcript level did not significantly differ between the mated and virgin queens’ spermathecae (Figure 2B). We also measured the catalase, GTPX, Tf, and SOD1 protein levels in the spermathecal fluid of mated and virgin queens by using antibodies generated against these proteins recombinantly produced in baculovirus-infected insect cells (Figure 3A,B). Notably, these results revealed that the SOD1, catalase, GTPX, and Tf protein levels were significantly higher in the spermathecal fluid of mated queens than in the spermathecal fluid of virgin queens.

Because Tf, catalase, GTPX, and SOD1 were upregulated in the mated queens’ spermathecal fluid (Figure 3), we hypothesized that ROS and H_2_O_2_ levels must be relatively higher in the spermathecal fluid of mated queens than in that of virgin queens, and thus antioxidant enzyme activities must also show the same trend as that of ROS and H_2_O_2_ levels. Notably, ROS and H_2_O_2_ levels were significantly higher in the spermathecal fluid of the mated queens than in the spermathecal fluid of the virgin queens (Figure 4A,B). Moreover, the activities of antioxidant enzymes—such as SOD1, catalase, and peroxidase—were increased, paralleling the ROS and H_2_O_2_ levels, in the mated queens’ spermathecal fluid (Figure 5A–C). These results are consistent with the hypothesis that antioxidant enzyme activities are higher in the mated queen’s spermathecal fluid owing to increased ROS and H_2_O_2_ levels; this phenomenon, in turn, indicates that an increase in antioxidant enzyme activity is an antioxidant defense mechanism against ROS in the spermathecal fluid of mated queens [10].

As previously described, Tf, an iron-binding protein, is upregulated in the spermathecae and spermathecal fluid of mated queens as opposed to those of virgin queens. Therefore, we compared the level of iron in the spermathecal fluid of mated queens with that of the virgin queens. Notably, iron content—including Fe^2+^ and Fe^3+^ concentrations—was higher in the mated queens than in the virgin queens (Figure 6), indicating that the upregulation of Tf in the spermathecal fluid of the mated queens is likely due to an increase in the iron level. This result is consistent with the increased H_2_O_2_ level in the spermathecal fluid of the mated queens and suggests that the Tf upregulation is a protective response to the Fenton reaction, which involves the production of HO˙ from H_2_O_2_ via Fe^2+^-catalyzed reactions [17,18,19,20,21,22]. Thus, these results suggest that the increased levels of antioxidant proteins, including Tf, in the mated queen’s spermathecal fluid play protective roles against ROS. 

### 3.3. MRJPs Are Differentially Expressed in the Spermathecae of Mated Queens

Interestingly, we found that *MRJPs 1–9* are expressed in the mated and virgin queens’ spermathecae and that these genes are differentially expressed in the spermathecae of mated queens (Figure 7). Among *MRJPs*
*1–9*, the transcript levels of *MRJP1*, *MRJP4*, and *MRJP6* were highly upregulated in the spermathecae of mated queens (Figure 7). The higher transcript levels were consistent with higher protein levels in the spermathecal fluid of mated queens (Figure 8, Appendix A). These results indicate that MRJPs were upregulated in the spermathecal fluid of mated queens and in particular, the upregulated levels of MRJP1, MRJP4, and MRJP6 were much higher than those of the other MRJPs.

## 4. Discussion

In this study, we examined the sperm in the mated honeybee queen’s spermathecal fluid 24 h after mating and performed transcriptomic analysis of the spermathecae of mated and virgin queens. The aim of these analyses was to characterize the differential gene expression patterns in honeybee queen spermathecae after mating and assess changes in the levels of Tf and MRJPs in the spermathecal fluid of mated honeybee queens. The findings have helped broaden and strengthen our understanding of the molecular basis of spermathecal fluid proteins in the spermathecae of honeybee queens. We first evaluated differential expression of *Tf* and *MRJPs* in the spermathecae of mated queens by conducting an RNA-seq analysis of the spermathecae extracted from both mated and virgin queens. Although extensive studies on the molecular basis of sperm storage in the spermathecae and spermathecal fluid of *A. mellifera* queens have been conducted [2,10,13,14], little is known about the upregulation of Tf and MRJPs in the spermathecae and spermathecal fluid of mated queens. In this study, therefore, we focused on the expression profiles of Tf and MRJPs, which were differentially expressed in the spermathecae and spermathecal fluid of mated queens.

The energy metabolism of the sperm stored in the spermathecae of mated queens is essential for their survival. However, these sperm also simultaneously produce ROS, which may decrease their viability during their storage in the spermathecae of honeybee queens [10]. As a strategy to minimize the ROS-induced damage to sperm, the ROS-production levels are minimized through reduction of the metabolic rate during storage [15]. A second strategy is to use an oxygen-independent pathway, which may minimize the ROS-induced damage in the stored sperm [4]. Honeybee sperm can maximize ATP production through aerobic metabolism and carry out non-aerobic energy production via the glycolytic pathway [4]. Another strategy is to minimize ROS production by producing antioxidant proteins in the mated queen’s spermathecal fluid. Various studies have revealed that antioxidant enzymes are detected in higher quantities in the mated queen’s spermathecal fluid than in that of virgin queens and have thereby suggested that antioxidant proteins protect the stored sperm from ROS [1,2,8−10]. The mRNA levels of antioxidant protein genes, such as *catalase*, *thioredoxin 2*, and *thioredoxin reductase 1*, were upregulated in the mated queen’s spermatheca, whereas no difference was observed in the mRNA level of the *SOD1* gene between the spermathecae of mated and virgin queens [10]. Moreover, Tf was detected in the spermathecal fluid and seminal fluid of honeybee queens [2,33]. Here, we observed that the antioxidant protein genes, such as *catalase*, *GTPX*, and *Tf*, showed a two-fold higher expression in the spermathecae of mated queens than in that of virgin queens, whereas the *SOD1* mRNA levels were not significantly different; this observation for *SOD1* is consistent with the findings of a previous study [10]. Notably, our results revealed that antioxidant proteins—such as catalase, GTPX, and Tf—were detected in higher quantities in the mated queen’s spermathecal fluid than in that of the virgin queens. Additionally, enzyme activity assays and western blot analyses revealed that the spermathecal fluid of mated queens exhibits higher SOD1 levels than the spermathecal fluid of virgin queen, as observed in the case of SOD enzyme activity in a previous study [8]. These results were consistent with increased ROS and H_2_O_2_ levels in the mated queen’s spermathecal fluid. Of the abovementioned antioxidant proteins, Tf upregulation is required for decreasing the enhanced iron level in the spermathecal fluid of mated queens, which suggests that Tf plays a role in suppressing toxic hydroxyl radical production via the Fenton reaction between Fe^2+^ and H_2_O_2_ [17,18,19,20,21,22]. Thus, our data suggest that Tf, which is upregulated in the spermathecal fluid of mated queens, may be responsible for antioxidant defense against ROS during sperm storage in the spermathecae of honeybee queens. 

Energy metabolism is essential for sperm storage in the spermathecae of mated queens. Sperm energy metabolism is coupled with ATP production by the mitochondria, which are required for complete oxidation of substrates such as sugar, fat, and proteins [11]. The advantage of carbohydrate-based energy metabolism is the production of ATP in the absence of oxygen (anaerobic metabolism) [11]. Moreover, GAPDH expression has been shown to be significantly higher in the stored sperm [4]. MRJP8 and MRJP9 have been identified as spermathecal fluid proteins in honeybee queens [2]. In this study, we found that *MRJPs 1–9* are differentially expressed in the spermathecae of mated queens. Since MRJPs 1–7 are expressed in the hypopharyngeal gland and MRJPs 8 and 9 are expressed in the venom gland [34,35], it is somewhat surprising that MRJPs 1–9 are detected in the spermathecal fluid of honeybee queens. Notably, these results provide the first evidence that MRJPs, especially MRJP1, MRJP4, and MRJP6, are detected in higher quantities in the mated queen’s spermathecal fluid than in virgin queens. Considering the nutritional role of MRJPs in the queen bee diet [36,37], our results suggest that highly expressed MRJPs in the spermathecae of mated queens may serve as a protein source for energy metabolism of the stored sperm.

In contrast, previous studies have demonstrated that MRJPs exhibit antimicrobial and antioxidant activities [27,38,39,40,41]. Thus, the present study could not exclude the possibility that MRJPs may function as antimicrobial and antioxidant agents and a protein source in the spermathecal fluid of honeybee queens. Further studies should attempt to define the functional roles of MRJPs in the spermathecal fluid of honeybee queens.

Additionally, we performed RNA-seq analysis, RT-PCR, and western blotting to reveal the differential expression of Tf and MRJPs in the spermathecal fluid and spermathecae of honeybee queens after mating. However, RT-PCR and Western blot data showed some discrepancies relative to the RNA-seq data, which was also described in a transcriptomic analysis of the honeybee queen spermathecae [14]. Our study indicates that the expressions of Tf and MRJPs are upregulated to various levels in the spermathecal fluid and spermathecae of honeybee queens after mating, in comparison with the corresponding expression levels in virgin queens, which suggests that the differential expression of Tf and MRJPs in the spermathecae and spermathecal fluid of mated queens may be involved in sperm storage [2,14].

In conclusion, our data provide evidence for the upregulation of Tf and MRJPs in the spermathecal fluid of mated queens, indicating their role in antioxidant defense and energy provision for the sperm stored in the spermathecal fluid of honeybee queens. Together, our findings elucidate the molecular-level changes in Tf and MRJP expression in the spermathecae and spermathecal fluid of mated honeybee queens, providing novel insights into the antioxidant defense and energy metabolism for stored sperm in honeybee queens.

## Figures and Tables

**Figure 1 insects-12-00690-f001:**
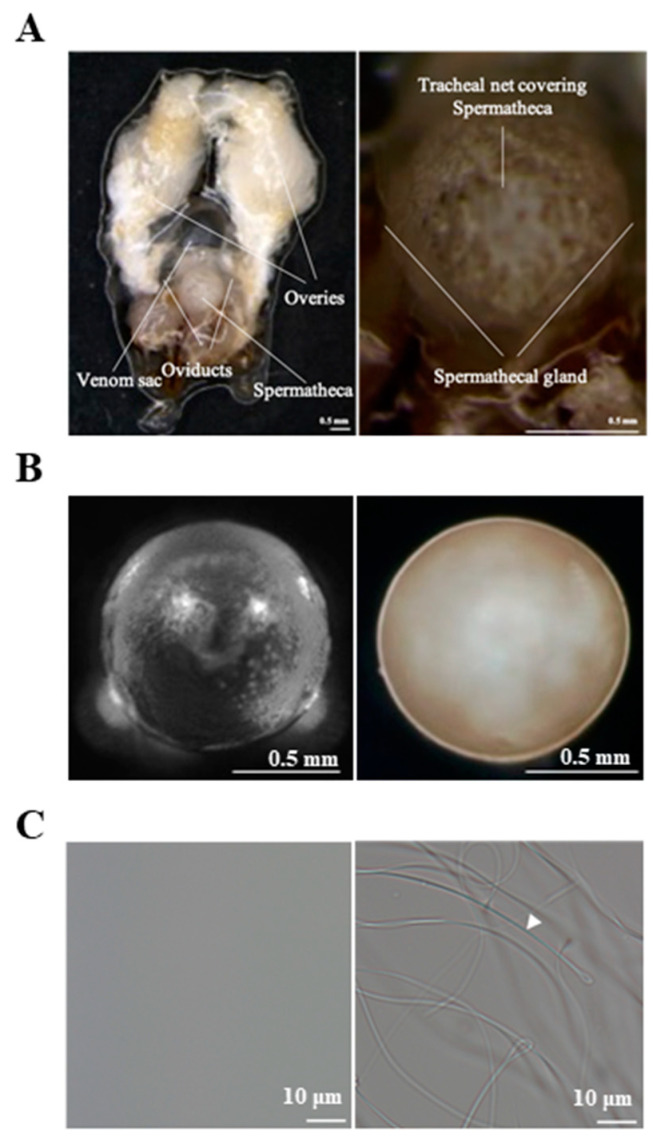
The reproductive organs, spermathecae, and sperm of *A. mellifera* queens. (**A**) The reproductive organs of the *A. mellifera* queen 24 h after mating. (**B**) The spermathecae of the virgin queen (Left) and the mated queen 24 h after mating (Right). (**C**) The spermathecal fluid of the virgin queen (Left) and the mated queen 24 h after mating (Right). The sperm in the spermathecal fluid of mated queens is indicated by arrowheads.

**Figure 2 insects-12-00690-f002:**
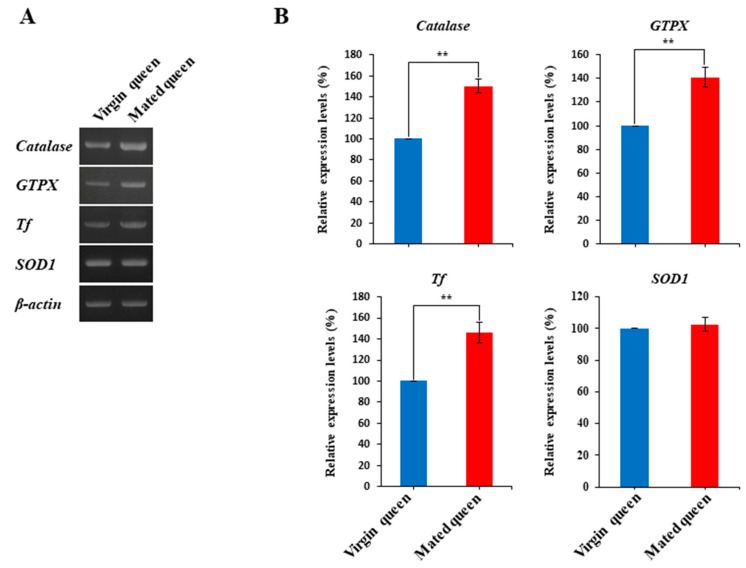
Transcript levels of genes encoding antioxidant proteins in the spermathecae of virgin and mated queens. (**A**) The transcripts of the antioxidant proteins *catalase*, *GTPX*, *Tf*, and *SOD1* in the spermathecae of virgin and mated queens were quantified using RT-PCR. *β-Actin* was used as the internal control. (**B**) The mRNA levels are shown relative to their levels in the spermathecae of virgin queens (shown as 100%). The bars represent the mean ± SD values. Significant differences (*p* < 0.01) are indicated with two asterisks.

**Figure 3 insects-12-00690-f003:**
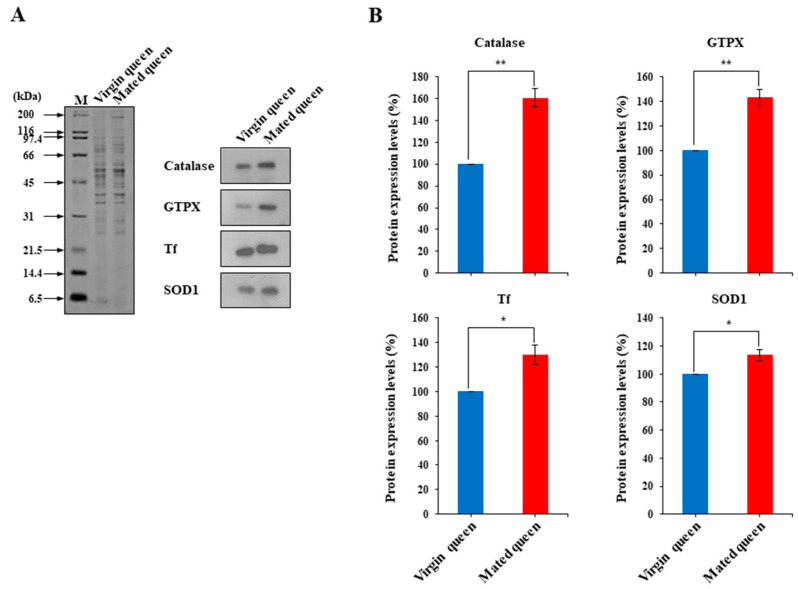
Detection of antioxidant proteins in the spermathecal fluid of virgin and mated queens. (**A**) Antioxidant proteins (catalase, GTPX, Tf, or SOD1) in the spermathecal fluid of virgin and mated queens were detected using SDS-PAGE (Left), and Western blot analysis was conducted using antibodies against catalase, GTPX, Tf, or SOD1 (Right). (**B**) The levels of catalase, GTPX, Tf, and SOD1 proteins are shown relative to their levels in the spermathecal fluid of virgin queens (shown as 100%). The bars represent the mean ± SD values. * *p* < 0.05 and ** *p* < 0.01.

**Figure 4 insects-12-00690-f004:**
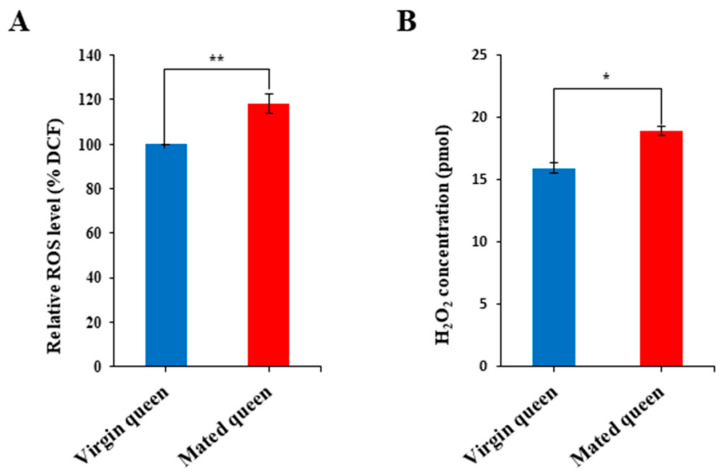
ROS (**A**) and H_2_O_2_ (**B**) levels in the spermathecal fluid of virgin and mated queens. The levels of ROS and H_2_O_2_ are shown relative to their levels in the spermathecal fluid of virgin queens (shown as 100%). The bars represent the mean ± SD values. * *p* < 0.05 and ** *p* < 0.01.

**Figure 5 insects-12-00690-f005:**
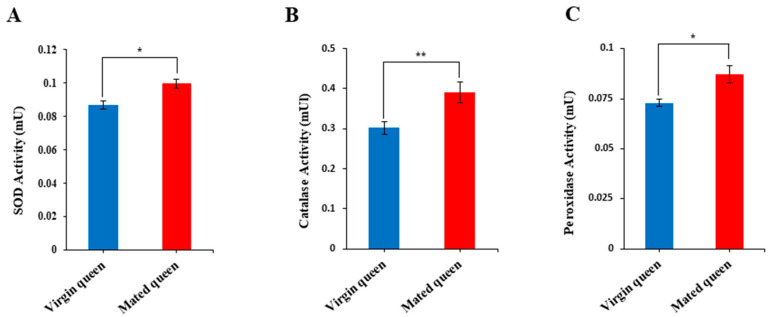
SOD (**A**), catalase (**B**), and peroxidase (**C**) activities in the spermathecal fluid of virgin and mated queens. The bars represent the mean ± SD values. * *p* < 0.05 and ** *p* < 0.01.

**Figure 6 insects-12-00690-f006:**
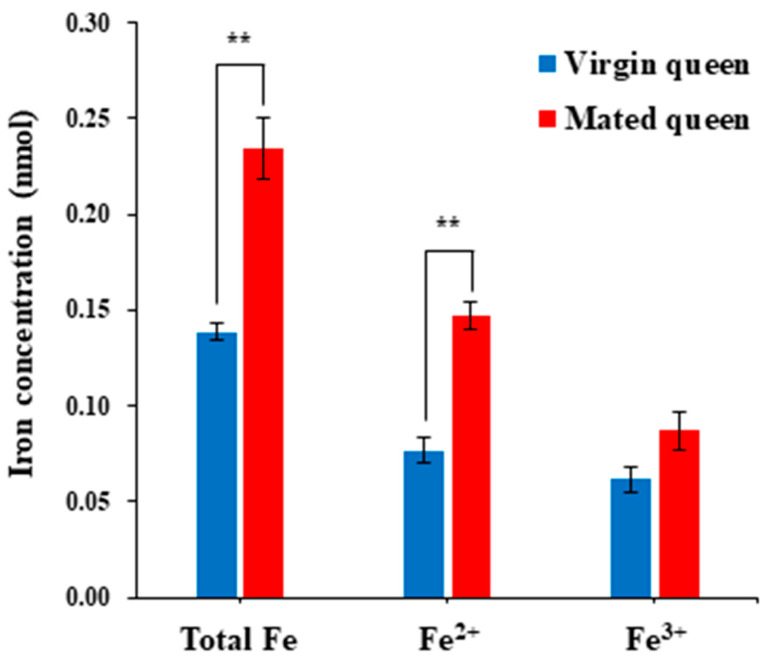
Iron concentrations in the spermathecal fluid of virgin and mated queens. The bars represent the mean ± SD values. Significant differences (*p* < 0.01) are indicated with two asterisks.

**Figure 7 insects-12-00690-f007:**
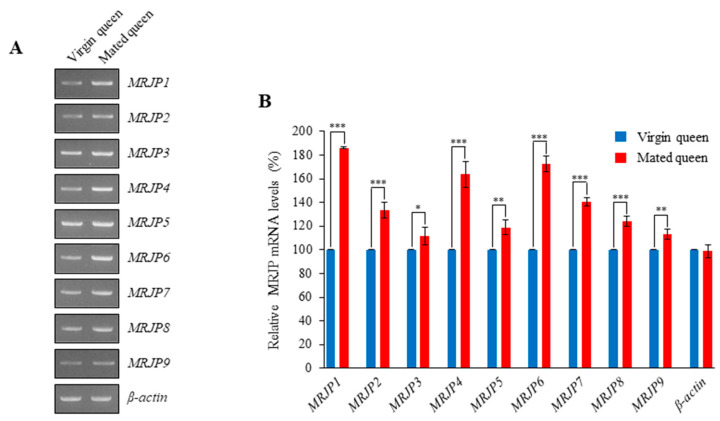
Transcript levels of *MRJP* genes in the spermathecae of virgin and mated queens. (**A**) The transcripts of *MRJP* genes (*MRJPs 1-9*) in the spermathecae of virgin and mated queens were quantitated using RT-PCR. *β-actin* was used as an internal control. (**B**) The *MRJP* mRNA levels are shown relative to their levels in the virgin queen’s spermathecae (shown as 100%). The bars represent the mean ± SD values. * *p* < 0.05, ** *p* < 0.01, and *** *p* < 0.001.

**Figure 8 insects-12-00690-f008:**
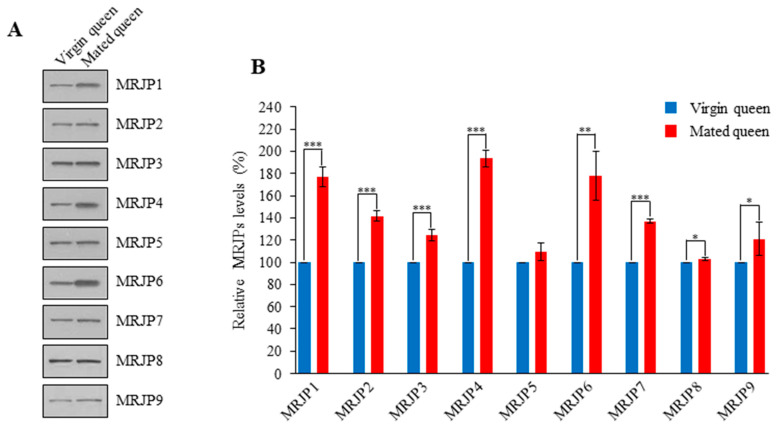
Detection of MRJPs in the spermathecal fluid of virgin and mated queens. (**A**) The MRJPs in the spermathecal fluid of virgin and mated queens were detected using western blot analysis conducted with antibodies against MRJPs. (**B**) The levels of MRJPs are shown relative to their levels in the spermathecal fluid of virgin queens (shown as 100%). The bars represent the mean ± SD values. * *p* < 0.05, ** *p* < 0.01, and *** *p* < 0.001.

**Table 1 insects-12-00690-t001:** Primers used in the RT-PCR analysis of the genes encoding antioxidant proteins and MRJPs in the spermathecae of virgin and mated queens.

Gene	Primer Sequence	Product Length (bp)
*Catalase*	forward 5ʹ-ATGACTGAAATAAAACGA-3ʹ	1542
	reverse 5ʹ-TCACAATCTAGCTGTCTTGCCATAA-3ʹ	
*GTPX*	forward 5′-ATGAGTGGAAACGACAAC-3′	507
	reverse 5′-TTAAAAATATTTTTCAAGATGATTT-3′	
*Tf*	forward 5′-ATGATGCTCCGATGCAAT-3′	2139
	reverse 5′-TTAAGCAGCACCGTAATTATTT-3′	
*SOD1*	forward 5′-ATGACTAAAGCAGTGTGC-3′	459
	reverse 5′-TTAGACTTTTGTAATTCCAA-3′	
*MRJP1*	forward 5′-ATGACAAGATTGTTTATG-3′	1299
	reverse 5′-TTACAAATGGATTGAAATTTTGAAA-3′	
*MRJP2*	forward 5′-ATGACAAGGTGGTTGTTC-3′	1359
	reverse 5′-TTAATTATCATTCTGATTGTTATTC-3′	
*MRJP3*	forward 5′-ATGACAAAGTGGTTGTTG-3′	1635
	reverse 5′-TTAATGTAATTTTGAAGAATGATGA-3′	
*MRJP4*	forward 5′-ATGACAAAATGGTTGCTG-3′	1395
	reverse 5′-TTAATCGTTATTGTTATGCCGATTG-3′	
*MRJP5*	forward 5′-ATGACAACTTGGTTGTTG-3′	1797
	reverse 5′-TTAATTATTATGCTTATTTTGATTG-3′	
*MRJP6*	forward 5′-ATGACAAATTGGTTACTG-3′	1314
	reverse 5′-CTAATCTAAATGAGCTTGATTCTTA-3′	
*MRJP7*	forward 5′-ATGACAAGGTGGTTGTTT-3′	1332
	reverse 5′-CTAATTAAATGAAGCGTCTATTGTG-3′	
*MRJP8*	forward 5′-ATGATAAGATGGTTGCTG-3′	1251
	reverse 5′-TCAAGGAGATATGCAACGAGTATTC-3′	
*MRJP9*	forward 5′-ATGTCTTTCAATATCTGG-3′	1272
	reverse 5′-TCAAAGGAAAATTGAGAAAAAATTT-3′	
*β–Actin*	forward 5′-ATGTCTGACGAAGAAGTT-3′	1131
	reverse 5′-TTAGAAGCACTTCCTGTGGA-3′	

**Table 2 insects-12-00690-t002:** Antioxidant proteins and MRJPs differentially expressed in the spermathecae of mated queens.

Protein	Gene ID	Fold Change
		Mated Queen/Virgin Queen
Tf	GB50226	2.51
Catalase	GB41427	2.77
GTPX	GB47478	2.07
MRJP1	GB55205	2.78
MRJP4	GB55206	3.00
MRJP6	GB55207	4.42

## Data Availability

Excluded.

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
