# Peer review of "Upregulation of Transferrin and Major Royal Jelly Proteins in the Spermathecal Fluid of Mated Honeybee (Apis mellifera) Queens"

_insects, 2021, doi:10.3390/insects12080690_

Round 1

Reviewer 1 Report

This is an interesting paper aiming at understanding the mechanism underlying long-term storage and survival of sperm in honeybee (Apis mellifera). The study revealed the upregulation of transferrin and major royal jelly proteins in the spermatheca and spermathecal fluid of mated honeybee queens using RNA-seq, RT-PCR, and western blot analyses. The authors provided a novel insight into the antioxidant defence and energy metabolism for the stored sperm of honeybee queens.

The introduction is detailed, takes into consideration the existing literature and adequately introduces to the topic of the study.

Materials & Methods have been properly detailed. However, a sentence at the beginning of the “section 2.1. Sample preparation” needs to be checked and rephrased.

The results have been properly presented with appropriate description, tables, graphs and pictures in order to make information clearly accessible and understandable.

Discussion:

The sentence on line 10 of the discussion needs to be revised.

Furthermore, the results have been adequately discussed and commented based on previous investigations and the existing literature.

Minor revisions are included in the text.

Author Response

The response  has been attached.

Reviewer 2 Report

The manuscript entitled “Upregulation of transferrin and major royal jelly proteins in the spermathecal fluid of mated honeybee (Apis mellifera) queens” is an interesting paper that applies different techniques to add pieces to the understanding of the mechanism underlying long-term sperm storage. There are however some points that need to be improved.

I divided my revision by the different sections of the paper.

Major concerns

Some methodological details are missing. The figures are very hard or even impossible to see. Some information, for instance, the MRJPs that are upregulated looks contradictory in different parts of the manuscript.

Specific concerns

Simple Summary, abstract and introduction

The authors wrote “confirm the upregulation of Tf and MRJPs…” and “our finding indicate that Tf and MRJPs are upregulated” However if I understand well not all MRJPs were upregulated, but only the MRJPs 1, 4, and 6 (see section 3.1, table 2 and section 3.3). The sentences should be changed accordingly.

The authors should also revise the same problem in the abstract and introduction

Abstract

The sentence that starts with “The concentration of Tf and antioxidant proteins…”and ends with “ of mated queen” is a bit long. It would be good if it could be divided.

Introduction

“We revealed that as opposed to the spermathecae of virgin queens, the spermathecae of mated queens showed upregulation of antioxidant proteins and Tf” which antioxidant proteins? If possible, they should be specified.

Material and Methods

In material and methods should be written how many samples/bees were used. This information is never provided and should be provided generally (how many bees are “all the bees”?) and for each step/method used here.

I do not understand the sentence “Spermathecae from 6-day-old virgin queens after

emergence or 7-day-old mated queens were kept for 24 h after artificial insemination using 6-day-old virgin queens were dissected on ice after emergence and artificial insemination, respectively using a stereomicroscope (Zeiss, Jena, Germany).” I believe that there are two different kinds of information here (1) how was performed the artificial insemination (2) how queens were dissected. But the sentence should be rewritten, maybe in two different sentences.

RNA Sequencing and Data analysis

What does mean that “RNA samples were prepared in triplicate n=3 spermathecae per replicate”. Maybe I do not understand that because the details of the number of samples are not provided. But also what is a replicate here?

Did the authors use any software for “For the analysis of differentially expressed genes (DEGs), log2 values were calculated and normalized through quantile normalization. Fold changes were calculated as the mean log2 (FPKM+1) value” If yes, they should tell the software.

Why the RNA reads were aligned to the reference genome Amel_4.5 and not to the new one Amel_HAv3.1?

Reverse Transcription-PCR (RT-PCR)

The authors tell the GenBank accession number for all genes but the MRJPs. However, they should also be provided. A good way of doing that is by adding all of them to Table 1.

The primers for the MRJPs were described in 25 and 26. And about the other primers that are also in table 1? Were they designed specifically for this study? If yes, the authors should provide how the primers were selected, which software was used.

Why the B-Actin and not other genes were used as internal control? The authors should add this information.

More details about RT-PCR should be provided, for instance, which thermocycler  was used?

How did the authors examine the expression patterns? did they use Ct values? Standard curves?

The results were analyzed by sequencing. This is very vague. Which kind of sequencing? Which machine? How and for what they were analysed?

Recombinant Protein Expression and Purification

Why the MRJPs genes were not used?

Statastical Analysis

Why different P-values were used? For different experiments? This should be further explained

Results

Identification of Genes Differentially Expressed between the Spermathecae of Mated and Virgin Queens

The legend of Figure 1 should be rewritten. In the first sentence looks that all the organs/tissues are from “…queens mated 24h after mating” after we understand that this information is not correct.  

More information can be provided and some other analysis can be performed. Please see below:

How many genes were not differentially expressed?

Why not do GO analysis with these 749 genes? Maybe different and interesting genes or family genes come out from this analysis. See for instance DavidDB which contains information for Apis mellifera.

A list of these 759 genes should be provided in supplementary data.

Why these specific antioxidant genes and not other are in table 2? In the 749 genes. only these antioxidant genes are differentially expressed?

In the way that the sentence is written, I was expecting that SOD1 was in Table2 but is not. The text should be clarified.

The gene abbreviation should be added to table 2 (eg Transferrin(Tf))

Antioxidant Defense in the Spermathecae of Mated Queens and Virgin Queens

“some antioxidant proteins” some is too vague. How many proteins?

“were higher in the spermathecal fluid of the mated queens than in the virgin” Is that significant?

Figure 2, 3, 4, 5, 6 and 7 are very small and blurry. Very difficult/impossible to see it carefully

“These results are consistent with the hypothesis that antioxidant enzyme activities

are higher in the spermathecal fluid of mated queens owing to increased ROS and H2O2

levels; this phenomenon, in turn, indicates that an increase in antioxidant enzyme activity is an antioxidant defense mechanism against ROS in the spermathecal fluid of mated queens” is it possible to add references?

MRJPs Are Differentially Expressed in the Spermatheca of Mated Queens

In the legend of table 2 says that only MRJP1, 4 and 6 are differentially expressed, but here does not say that.

Author Response

Thanks for your kind and valuable comments, we have attached the response point by point

Round 2

Reviewer 2 Report

In my opinion, generally the authors of the manuscript entitled “Upregulation of transferrin and major royal jelly proteins in the spermathecal fluid of mated honeybee (Apis mellifera) queens” answered my questions. However, some of them were not completely addressed. 

1)Just to be sure, did the authors used 3 virgin queens and 3 mated queens? Is that correct?
2)were the primers listed in table 1 designed for this manuscript? If yes, a small description of how they were designed should be added. For instance, which software did the authors use? You can see for instance how Soares, S., Grazina, L., Mafra, I., Costa, J., Pinto, M. A., Oliveira, M. B. P., & Amaral, J. S. (2019). Towards honey authentication: Differentiation of Apis mellifera subspecies in European honeys based on mitochondrial DNA markers. Food chemistry, 283, 294-301. Described the primer design. 
 3) Following my suggestion the authors described in the methods section that they performed the DAVIDB analysis. However, I did not find the results of this analysis.
4) I understand the author’s answer concerning the gene list of the 759 genes. However, I think that there is no problem if the 759 genes found here are similar to the genes found by Rangel et al 2021. Actually, I think that this gives even more strength to the work. And as supplementary data, it does not “occupy” so much space. However, maybe the editor should decide if the authors should provide the gene list.

Author Response

Ms No.: Insects-1301011-R1

In my opinion, generally the authors of the manuscript entitled “Upregulation of transferrin and major royal jelly proteins in the spermathecal fluid of mated honeybee (Apis mellifera) queens” answered my questions. However, some of them were not completely addressed. 

Response: We are grateful for the kind and valuable comments. According to Reviewer 2’s comments, we revised.

1) Just to be sure, did the authors used 3 virgin queens and 3 mated queens? Is that correct?

Response: We revised the Sentences according to the comment of the reviewer. 9 samples of virgin queens or mated queens were used.

2) were the primers listed in table 1 designed for this manuscript? If yes, a small description of how they were designed should be added. For instance, which software did the authors use? You can see for instance how Soares, S., Grazina, L., Mafra, I., Costa, J., Pinto, M. A., Oliveira, M. B. P., & Amaral, J. S. (2019). Towards honey authentication: Differentiation of Apis mellifera subspecies in European honeys based on mitochondrial DNA markers. Food chemistry, 283, 294-301. Described the primer design. 

Response: The PCR primers for MRJPs 1–9 have been described in our previous studies [27,28]. Additionally we designed the primers to amplify full sequence including start to stop codons.

 3) Following my suggestion the authors described in the methods section that they performed the DAVIDB analysis. However, I did not find the results of this analysis.

Response: Thank you very much for kind comment on that. According to Reviewer 2’s comments, we add it as Table S2.

4) I understand the author’s answer concerning the gene list of the 759 genes. However, I think that there is no problem if the 759 genes found here are similar to the genes found by Rangel et al 2021. Actually, I think that this gives even more strength to the work. And as supplementary data, it does not “occupy” so much space. However, maybe the editor should decide if the authors should provide the gene list.

Response: Thank you very much for kind comment on that. According to Reviewer 2’s comments, we provide it as Table S1.